# Concurrently Fabricating Precision Meso- and Microscale Cross-Scale Arrayed Metal Features and Components by Using Wire-Anode Scanning Electroforming Technique

**DOI:** 10.3390/mi14050979

**Published:** 2023-04-29

**Authors:** Shicheng Li, Pingmei Ming, Junzhong Zhang, Yunyan Zhang, Liang Yan

**Affiliations:** School of Mechanical and Power Engineering, Henan Polytechnic University, Jiaozuo 454000, China

**Keywords:** cross-scale arrayed metal, thickness uniformity, wire-anode scanning electroforming

## Abstract

In order to improve the thickness uniformity of the electroformed metal layer and components, a new electroforming technique is proposed—wire-anode scanning electroforming (WAS-EF). WAS-EF uses an ultrafine inert anode so that the interelectrode voltage/current is superimposed upon a very narrow ribbon-shaped area at the cathode, thus ensuring better localization of the electric field. The anode of WAS-EF is in constant motion, which reduces the effect of the current edge effect. The stirring paddle of WAS-EF can affect the fluid flow in the microstructure, and improve the mass transfer effect inside the structure. The simulation results show that, when the depth-to-width ratio decreases from 1 to 0.23, the depth of fluid flow in the microstructure can increase from 30% to 100%. Experimental results show that. Compared with the traditional electroforming method, the single metal feature and arrayed metal components prepared by WAS-EF are respectively improved by 15.5% and 11.4%.

## 1. Introduction

Length scales for manufacturing processes are generally classified as nanoscale (<100 nm), microscale (100 nm to 100 μm), mesoscale (0.1 mm to 10 mm), and macroscale (>0.5 mm) [1]. It can be seen that the meso-manufacturing process is cross-scale manufacturing; it fills the gap between macro- and micromanufacturing processes and overlaps both of them. At each scale, of course, there are various available manufacturing processes based on different governing mechanisms developed for a variety of engineering materials and geometric shapes. In the miniature engineering fields, a wide variety of microscale and mesoscale metal (M^3^) features and components with high accuracy and precision are highly demanded, for example, mechanical watch gears [2], small motors and bearings, mesoscale fuel cells [3], micro-/meso-scale pumps and valves [4,5], medical stents [6], mini nozzles, printing stencils for electronic packaging, and many others.

Manufacturing at the microscale and mesoscale can be accomplished by scaling down macroscale manufacturing processes or scaling up nanomanufacturing or micromanufacturing processes. The scaling down processes create the features and components by subtractively removing materials gradually; hence, they are also called subtractive manufacturing processes. Oppositely, the latter are also called additive manufacturing (AM) processes. The subtractive micro and mesoscale metal manufacturing (subtractive-M^4^) processes generally produce the features and components from the given materials one by one, and include mechanical machining (milling [7], turning, drilling [8], etc.), laser beam machining [9], focused ion beam machining [10,11], electro-discharge machining [12], and chemical/electrochemical machining [13,14], etc. The additive-M^4^ processes produce the objects by depositing materials typically layer by layer and include electrochemical deposition (ECD) [15], electroless plating [16,17], focused-ion beam induced deposition (FIBID) [18], laser-chemical vapor deposition (LCVD) [19], etc. The additive-M^4^ processes are increasingly attractive since they can flexibly create geometric shapes and controllably tailor the material performances synchronously.

ECD is a widely used method to prepare metal materials with different forms such as film, coating, powder, bulk, and articles in a wide range of industrial applications, and correspondingly it has different application modes including electroplating, electrolytic refining, and electroforming. Compared with other additive-M^4^ processes, ECD is more competitive in cost-efficiency, machinable materials, geometric shapes, and sizes, forming accuracy and precision, material compactness, etc. More attractively, it enables the manufacture of M^3^ features and components in a large volume simultaneously in the same electrochemical reaction system with the photoresist mask-based electrodeposition (mask-based ECD) processes. The TM-ECD (through-mask ECD) forms precision M3 features and components by depositing the reduced metal atoms into the accurately pre-patterned thick-photoresist mask cavities which are finally removed, and so it is generally considered as a photoresist through-mask defined process, as shown in Figure 1. Currently, the TM-ECD has been preferably employed to manufacture M^3^ features and components used for various applications.

In the traditional TM-ECD processes, electrodeposition is generally carried out by placing the cathode and anode to completely immerse in the electrolyte bath with the cathode and anode facing each other. The distance between the two electrodes is usually required to be not less than 10 cm to allow necessary mass transfer activities and related operations. Furthermore, to maintain normal dissolution of the anode, the area ratio of the cathode to the anode is generally greater than 1.5. These requirements mentioned above inevitably cause some problems, in which the uneven distribution of the deposit thickness in different through-mask cavities is the most frequently observed problem. The unevenness of the thickness distribution during the traditional TM-ECD stems from the nonuniform distribution of the current density at different electrodeposition areas, and in turn, the nonuniform electric field is mainly caused by the existence of the interelectrode gap (IEG) and through-mask patterns at the cathode as well as by the difference in the mass transportation rate at the different place. That is to say, the distribution characteristics of the mass transfer field and electric current field are the fundamental factors determining the deposit thickness distribution during TM-ECD. The uneven distribution of the deposit thickness will result in a significantly reduced dimensional accuracy even the failure of the deposited M^3^ features and components. Therefore, a number of investigators have made a variety of efforts to improve the deposit thickness distribution during TM-ECD.In the traditional maskless electrodeposition processes, assisting auxiliary electrode/thief electrode (cathode or anode) [20,21,22,23,24], using an electric field shield/conformal anode [25], reciprocating the cathode [26], applying modulated electric current [27,28], and adding appropriate additives [29,30,31] are frequently used methods to improve the deposit thickness distribution. However, these traditional methods are less effective for the TM-ECD processes, since the patterned photoresist through-masks are inherently inhomogeneous regarding the distribution of the current density and also the mass transfer rate distribution, especially when the electrodeposition is carried out via electrically filling the multiple-scale through-masks. Therefore, some investigators made a great effort to additionally explore some nontraditional methods to further homogenize the patterned deposits.

Tsai et al. [32] carried out the electroforming process in a highly pressurized bath to uniformize the deposit thickness. Zhai et al. [33] reported that appropriately applying ultrasonic or megasonic field to the bath solution can uniformize electrodeposition rate distribution during through-mask electroforming. Jie et al. [34] achieved an improved thickness uniformity by electrodepositing metals from the supercritical solution. Some researchers verified that the use of the reciprocating-paddle agitation close to the top surface of the patterned through-mask is significantly beneficial to achieve an improved deposit thickness.

This paper proposes a unique TM-ECD that is fundamentally different from the traditional TM-ECD processes in which a large surface-area ratio between the anode and cathode is used and the IEG is also very large. In the proposed TM-ECD, a linear wire-shaped ultrafine (diameter/width ≤ 50 µm) electrically inert anode (abbreviated as wire-anode) that is fixed on a reciprocating paddle is used, and the working gap between the anode and the top surface of the through-mask is kept below 500 µm. Here, this novel TM-ECD is especially called wire-anode scanning electroforming (WAS-EF). During WAS-EF, the stirring paddle is moving reciprocally with a constant speed. With WAS-EF, it is expected to achieve high-precision deposition of a variety of multi-scale components on the same cathode at the same time. In the following sections, verification studies will be carried out theoretically and experimentally with various evaluations including electrodeposition behaviors, surface morphology, dimensional accuracy, etc.

## 2. Working Scheme of Wire-Anode Scanning Electroforming

Figure 2 shows the schematic diagram of the WAS-EF. The electric potential/current from a static anode superimposes a very narrow ribbon-shaped area at the cathode, showing a Gaussian distribution characteristic. It was found in that, depending on the working gap between the anode and the top surface of the through-mask, the ribbon width is generally less than 100 times as wide as the width/diameter of the anode. The theoretical model of a static WSA-EF is shown in Figure 2b. Based on the Butler–Volmer equation, the overpotential deriving from the wire-anode at the cathode also shows a similar Gaussian distribution characteristic. The overpotential distribution function, η(w), can be approximately expressed as follows [35],(1)$$\eta \left(w\right)=\frac{2\eta_{0}}{\pi w_{0}^{2}}\exp\left(-\frac{2w^{2}}{w_{0}^{2}}\right)$$
where w is the horizontal distance away from the anode’s central line, η_0_ is the overpotential value when w = 0, and w_0_ is the distance away from the anode central line when η(w_0_) = η_e_ (η_e_ is the critical evolution overpotential of a specific metal ion).

According to the metal growth mechanism during WAS-EF, the resulting electrodeposited material can be divided horizontally into m micro-layers, and each microlayer can be divided vertically into n micro-strips, as shown in Figure 2b. The length dw of the micro-strip is calculated as follows,
(2)$$\mathrm{dw}=\frac{L}{n}$$
where L is the total length of electrodeposited material along the scanning direction. The deposition time of the micro-strip can be estimated by the following formula,
(3)$$t=\left(\mathrm{dw}+\left|2w_{0}\right|\right)/v$$
(4)Δm∝t
where v is the scanning speed of the wire-anode and Δm is the height of a single micro-layer.

According to Formulas (2)–(4), the deposit thickness is majorly controlled by the scanning speed. This is significantly different from the traditional bath electroplating processes in which the deposit thickness is affected by quite a few process parameters. Therefore, a very uniform deposit can be achieved if the deposition time is deliberately selected.

Figure 3 schematically shows the working mode when the WAS-EF is used to fabricate micro/mesoscale features and components in batches with through-masks. Unlike the blank WAS-EF with a constantly homogeneous electric current field, the mask-based WAS-EF may have a varying electric field distribution changing with the location change of the scanning anode. As shown in Figure 3b, during the scanning of the wire-anode, the electric current distribution varies when the anode approaches and is right over the through-masks. This is due to the existence of dielectric material (through-masks) which locally changes the electric conductivity. It is also shown that the electric current distribution varies somewhat with the size of the through-mask’s cavity during the scanning of the anode. However, these differences may be greatly reduced because the electric energy acts very locally and keeps moving, and especially, the mass transfer rate in the working gap stirred by a moving triangle-sectional paddle is little affected by the size of the cavities. It has been verified that, with the stirring of a triangular paddle moving slowly, laminar-flow convection mass transfer forms outside the mask’s cavities, but diffusion mass transfer dominates within the cavities. These beneficial factors may greatly improve the current distribution and thus the thickness distribution uniformity of WAS-EF.

## 3. Simulation of Electric and Flow Fields during WAS-EF

### 3.1. Modeling and Conditions

According to the machining principle described above, the wire-anode scanning electroforming machining is simplified into a physical model of numerical simulation, as shown in Figure 4. In the direction of calculating the length of the domain, three kinds of micro-groove structures with a thickness of 70 μm and width of 0.07 mm, 0.1 mm, 0.2 mm, and 0.3 mm were set. The wire-anode and the stirring paddle are located 100 μm above the through-mask. The COMSOL Multiphysics (5.5a) software was used to simulate the model. The secondary current distribution interface was selected in the simulation physical field, and the flow of electrolyte and the reciprocating linear motion of wire-anode were realized through the fluid-structure coupling interface. The flow field and electric field were coupled by the conductivity. To improve the convergence of simulation calculation, the mesh is automatically re-divided in the calculation. Data collection is performed during a single process of wire-anode movement. The corresponding time is as follows, S1-0.6s, S2-1.5s, S3-2.4s, S4-3.4s, S5-4.7s, S6-6.4s, S7-8.4s, S8-11s, and S9-13.7s.

The fluid flow model adopts the flow governing equation of incompressible fluid, namely, the Navier–Stokes equation [36], and the relevant governing equation is as follows.
∇u = 0(5)
(6)$$\rho (\frac{\partial u}{\partial t}+u\cdot \nabla u)=-\nabla p\;+\nabla (\mu (\nabla u\;+{\left(\nabla u\right)}^{T}))+\;F$$
where ρ stands for density (kg/m^3^), μ for dynamic viscosity (N·s/m^2^), u for velocity (m/s), p for pressure (Pa), and F for surface tension.

The secondary current density distribution was used to characterize the electrochemical kinetic reaction at the cathode interface, that is, the Bulter–Volmer equation [37] was used as the kinetic model of the electrochemical polarization reaction at the cathode.
(7)$$i_{\mathrm{loc}}\;=\;i_{0}\;(\exp(\frac{\alpha_{a}\;F\eta }{\mathrm{RT}})-\;\exp(\frac{\alpha_{c}\;F\eta }{\mathrm{RT}}))$$
η = ϕ_s_ − ϕ_l_ − E_eq_(8)
where i_loc_ is the local current density, i_0_ is the exchange current density, α_a_ is the anode reaction coefficient, α_c_ is the cathode reaction coefficient, η is the over-potential, F is the Faraday constant with the value of 98,485 C/mol, R is the gas constant with 8.314 J/(mol·K), T is the thermodynamic temperature, ϕ_s_ is the electrode potential, and ϕ_l_ is electrolyte potential; E_eq_ is the equilibrium potential of electrodeposited metal.

As shown in Figure 4b, domain Ω_1_ is the electrolyte, domain Ω_2_ is the stirring paddle, boundary 2 is the electrolyte inlet, boundary 20 is the electrolyte outlet, boundary 26 is the wire-anode, boundary 3–19 is the photoresist, boundary 5, 9, 13, 17 is the cathode surface, and boundary 21 to 25 is the stirring paddle boundary. Other specific settings and parameters are shown in Table 1. Boundary conditions for numerical simulation of wire-anode scanning electroforming are shown in Table 2.

To simplify the calculation without losing generality, the following assumptions were made in terms of flow field: (1) the electrolyte is an incompressible viscous fluid, and the conductivity, temperature, density, and other parameters in the electrolyte remain unchanged and equal at each position; (2) The initial state of electrolyte is static, and the influence of electrolyte gravity is ignored; (3) The temperature does not change during the electrode reaction.

### 3.2. Electric Field Distribution in the Cathode Region

To better represent the current density distribution on the cathode surface in the process of wire-anode scanning electroforming, nine feature points were selected to analyze the current density distribution on the cathode surface based on a moving stroke.

As shown in Figure 5, the current density in the microstructure was the highest when the anode was directly above the microstructure (S2, S4, S6, S8), and the current density decreased as the anode moved away. The current density distribution is affected by the depth-to-width ratio. When the depth-to-width ratio is smaller, the current density per unit surface is lower, and the current distribution tends to be uniform with the increase in the aspect ratio.

Because the wire-anode is in constant motion, the current density distribution also changes in real time. Taking the microstructure IV as an example, when the wire-anode distance is far away from the microstructure IV (located at S1), the current density distribution is more uniform. When the wire-anode is in the S8 position (directly above IV), the current density distribution on the surface of microstructure IV is a Gaussian distribution, indicating that the width of the ribbon is less than the width of microstructure IV, which is consistent with what was said before. When the wire-anode is located at S7 and S9, the current density distribution on the cathode surface is reversed. In other words, the anode compensates for the current density distribution on the cathode surface during its motion. When the anode is on the left side of the area to be deposited, the high (low) current density area on the surface will be compensated when the anode is on the right side, thus achieving a dynamic equilibrium process.

The current density on the cathode surface of wire-anode scanning electroforming was superimposed (when the wire-anode was at nine characteristic points), and compared with the traditional electroforming mode, as shown in Figure 6. It can be seen that under the condition of wire-anode scanning electroforming, the current density distribution on the cathode surface is more uniform, which reduces the influence of bending electric field lines into the deposition area and improves the uniformity of the electroforming layer.

Therefore, compared with traditional electroforming methods, the wire-anode scanning electroforming method can improve the uniformity of electroforming current density to some extent.

### 3.3. Distribution of Flow Field in the Cathode Region

Figure 7 shows the influence of the stirring paddle on the cathode surface flow field when it is above the microstructure. It can be seen that the eddy generated by the stirring paddle motion will affect the fluid flow in the microstructure, which will improve the mass transfer effect inside the structure.

Figure 8a shows the flow velocity at the center of each microstructure of the stirring paddle under a static state. It can be seen that, due to the small size of the micro-structure, the internal influence of the flow field is weak. Figure 8b shows the fluid velocity in the inner central region of each microstructure when the stirring paddle is located above each structure (S2, S4, S6, S8).

It can be seen that the agitation effect of the stirring paddle on the electrolyte in the microstructure becomes more and more obvious with the decrease in the aspect ratio. When the depth-to-width ratio decreases from 1(Ⅰ) to 0.23(Ⅳ), the agitation caused by the stirring paddle can range from 20 μm to 70 μm. In this way, the mass transfer mode of electroforming is determined by diffusion and forced convection.

## 4. Experimental Study

### 4.1. Experimental Setup and Materials

The setup developed for WAS-EF is schematically shown in Figure 9. The electrolyte solution tank consists of the electroforming main tank and the electrolyte reservoir. An overflowing plate separating the electroforming tank and electrolyte reservoir was used to maintain and regulate the electrolyte height. The isosceles triangle cross-section of the used stirring paddle is 40 mm (B) and × 70 mm (S). The paddle is made of polymethyl methacrylate polypropylene material. The anode is made of the platinum strip (99.999 wt%, 100 mm × 20 mm × 0.05 mm) which is embedded in the paddle with their bottom surfaces aligning with each other. The reciprocating scanning speed of the wire-anode is 1.0–100 mm/s. The working gap can be regulated, ranging from 10 μm to 70 μm. The wire-anode travel is 0–150 mm.

The polished SUS304 stainless steel plate (100 mm × 100 mm × 1 mm) was used as the cathode substrate and was carefully pre-treated before its use. The photoresist through-masks were subsequentially prepared on the cathode substrates by using the standard dry photoresist film ((Dupont, Wilmington, DE, USA) lithographical processes. The thickness of the through-masks is 70 ± 1 μm, and the details of the through-masks are shown in Figure 10.

The electrolyte compositions were nickel sulfamate (500 g/L, AR, 98%) and boric acid (15 g/L, AR, 98%). Because of the use of the electrochemically inert anode, nickel chloride was not added. The electrolyte temperature was kept at 50 ± 1 °C. The pH value of the electrolyte was maintained at 4 ± 0.2 by adding sulfamate acid or nickel carbonate. A composition online testing and the auto-adding system were used to maintain compositions in the electrolyte at a designated value. A direct-current power supply (ITECH, IT6122B, China) was applied to the electroforming processes, and the working voltage was controlled at 2.6–6.0 V.

### 4.2. Surface Morphologies and Geometrical Dimension Accuracy Characterization

Observation of the surface morphology and topographies of the fabricated features and structures was conducted with a scanning electron microscope (Merlin Compact, Carl Zeiss NTS GmbH, GER), and laser confocal microscopes (Olympus, OLS5100, JP).

To facilitate the characterization of the thickness distribution of the samples with different scales, two parameters, θ, and η, were defined. θ was defined as the thickness distribution uniformity of the single-scale features, and η was defined as the thickness distribution uniformity of multiscale combined features, as shown in Figure 11.
(9)$$\theta =\left(1-\frac{D_{\max}-d_{\min}}{d_{m}}\right)\times 100\%$$
(10)$$d_{m}=\frac{D_{\max}+d_{\min}}{2}$$
(11)$$\eta =\left(1-\frac{H_{\max}-h_{\min}}{h_{m}}\right)\times 100\%$$
(12)$$h_{m}=\frac{\sum_{i=1}^{n}\left(H_{i}+h_{i}\right)}{n}$$
where D_max_ and d_min_, respectively are the maximum and minimum thickness of the single part, d_m_ is the average thickness of the single part, H_max_ and h_min_ are the maximum and minimum thickness of i parts in the part array, h_m_ is the average thickness of all electroformed components using the same through-mask, and n is the number of electroformed parts using the same through-mask.

## 5. Results and Discussion

### 5.1. Surface Morphology Analysis of Parts

Figure 12 shows the effect on surface quality at different machining voltages, from 2.6 to 6 V. When the electroforming voltage is low (2.6 V), the surface of the parts is not smooth, and there are local protrusions and other defects, as shown in Figure 12a. This is because under the condition of low voltage, the reaction is slow and the bubbles are generated slowly. Therefore, large bubbles cannot be quickly formed and be desorbed on the electrode surface, resulting in bubble accumulation and resulting in poor surface quality. When the machining voltage is 4 V, the surface morphology of the electrodeposited layer is smooth, as shown in Figure 12b. When the voltage is too high (6 V), there are more nodules on the surface of the parts, as shown in Figure 12c. The parts made by traditional electroforming are shown in Figure 12d; under the same conditions, the surface of the electroforming is poor, and there is insufficient deposition and other phenomena. This is because the accumulation of bubbles in the electroforming process will block the photoresist surface and thus affect the electroforming process. However, in the wire-anode scanning electroforming mode, the reciprocating movement of the stirring paddle will accelerate the desorption of bubbles from the electrode surface, thus reducing the influence of bubbles on the electroforming process.

Figure 13 shows the structural characteristics of the parts under different working gaps. As the working gap increases, the thickness uniformity of the parts decreases, which is because the parts are affected by the current edge effect. As mentioned above, the width of the current distribution at the cathode of wire-anode scanning electroforming depends on the size of the working gap. Figure 13e–h shows a schematic diagram of the current distribution in line anode scanning electroforming. When the working gap is small, the current distribution is narrow, the action area is small and hindered by the film, and the edge effect of the current is small. When the working gap is large, the action area becomes wider, when the wire-anode is located at the edge, the through-mask on the power line obstruction effect is weak, the influence of the current edge effect is larger, and the electrodeposited metal will become uneven.

### 5.2. Characteristic Analysis of Single Part

Figure 14a–c shows the thickness characteristics of different sizes of parts under the same electroforming condition. It can be seen that, when the processing conditions are the same, the thickness of larger parts is significantly lower than small ones. This is because the same current density is applied to larger machining areas, and the average current density is lower, so the deposition rate is smaller and the deposition thickness is smaller; this is consistent with the current density distribution shown in Figure 5. Microcomponents with sizes of 100 μm and 1500 μm show a saddle shape with both sides high and the middle low, which is still affected by the edge effect of the current. When the microcomponent with the size of 300 μm presents a low convex shape on both sides of the middle process, this is because the mass transfer of the electrolyte on the side wall of the mask is constrained by the surface tension effect, so the electrolyte flow rate in the middle of the pit is faster than that on the side wall, and the final surface profile of the electrodeposited part is convex. In contrast, microstructures of the same size were prepared by the traditional electroforming method, as shown in Figure 14d–f. It can be seen that the influence of the current edge effect is more serious, and the thickness of the electroforming layer is more uneven.

### 5.3. Characteristic Analysis of Part Arrays

In the actual electroforming process, electroforming parts often face the problem of multi-scale/cross-scale, which is also the key factor affecting the thickness uniformity of parts at multi-scale/cross-scale. Therefore, the film containing pattern arrays were prepared on the substrate and used the optimized wire-anode scanning electroforming parameters to process

Figure 15 show the characteristics of the macro and micro parts of wire-anode scanning electroforming and traditional electroforming. It can be seen that the thickness uniformity of wire-anode scanning electroforming is better than that of traditional groove plating. However, from the aspect of scale effect characteristics (concave and convex on the surface of parts), the wire-anode scanning electroforming and traditional electroforming parts are affected by the edge effect, and their profiles cannot be completely flat. This is because of the existence of the photoresist; the substrate is divided into several conductive and non-conductive regions with different areas, and the obstruction and interference of the film on the electric field line will result in more chaotic and distorted current distribution on the cathode surface. Therefore, the current density of each part of the deposition zone is biased, so the thickness will also be affected.

### 5.4. Parametric Analysis of Thickness Uniformity of Electroforming Parts

To evaluate the thickness uniformity of electroforming parts parametrically, θ and η defined above are used to evaluate them. Shown in Figure 16 are the thickness uniformity evaluation of single-size parts, and the uniformity evaluation of part arrays. Compared with the traditional electroforming method, the single metal feature and arrayed metal components prepared by WAS-EF are, respectively, improved by 15.5% and 11.4%. This shows that the effect of the current edge effect can be reduced by wire-anode scanning electroforming.

## 6. Conclusions

In this paper, to prepare components with better thickness uniformity, a novel WAS-EF is proposed, in which a wire-anode is used and is kept moving to and fro over the cathode with a markedly small working gap. The simulation mode for WAS-EF was established, and the working principle was analyzed. Some conclusions are made as follows.(1)Through numerical simulation, it can be concluded that the eddy current generated by a stirring paddle motion can affect the flow field in the micropore, resulting in a better mass transfer effect on the cathode surface. On the other hand, by stacking electric fields, it can be seen that the current density is higher and the distribution is more uniform in the mode of wire-anode scanning electroforming.(2)The results show that the uniformity of components decreases with the increase in the working gap. When the machining voltage is high or low, the surface quality of components is not ideal. Under the optimized working gap and machining voltage, components with better uniformity can be obtained.(3)Compared with the traditional electroforming technique, the thickness uniformity of wire-anode scanning electroforming is improved by 15.5% on average for the preparation of single-scale components, and 11.4% for the preparation of multi-scale component arrays.


## Figures and Tables

**Figure 1 micromachines-14-00979-f001:**
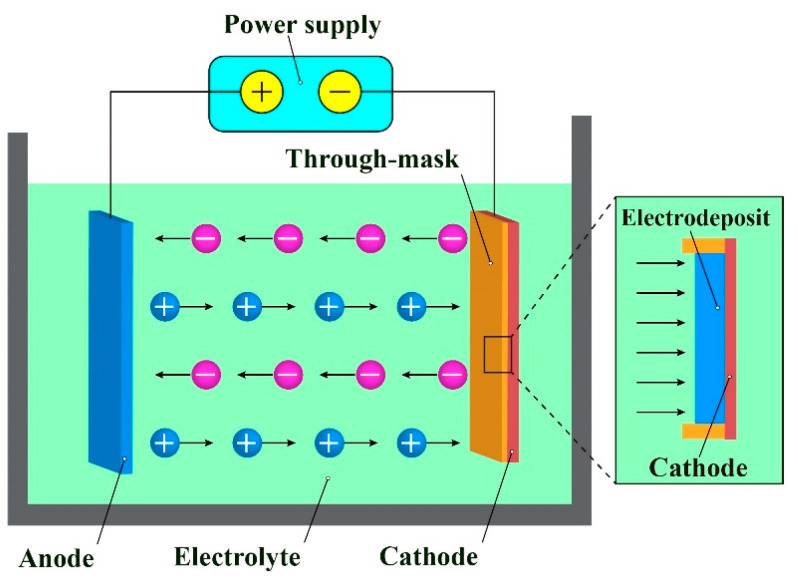
Working scheme and mechanism of TM-ECD.

**Figure 2 micromachines-14-00979-f002:**
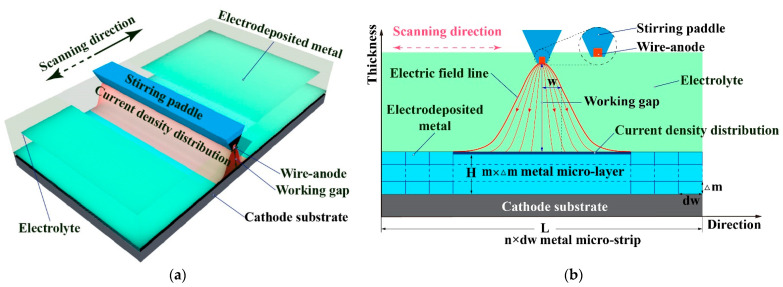
Schematic diagram of the WAS-EF without using through-masks and its forming mechanism. (**a**) schematic diagram of the WAS-EF; (**b**) forming mechanism of deposit during WAS-EF.

**Figure 3 micromachines-14-00979-f003:**
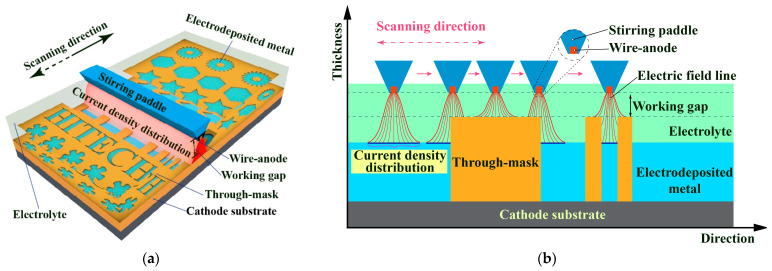
Schematic diagram of the WAS-EF using through-masks and its forming mechanism. (**a**) schematic diagram of the WAS-EF; (**b**) forming mechanism of deposit during WAS-EF.

**Figure 4 micromachines-14-00979-f004:**
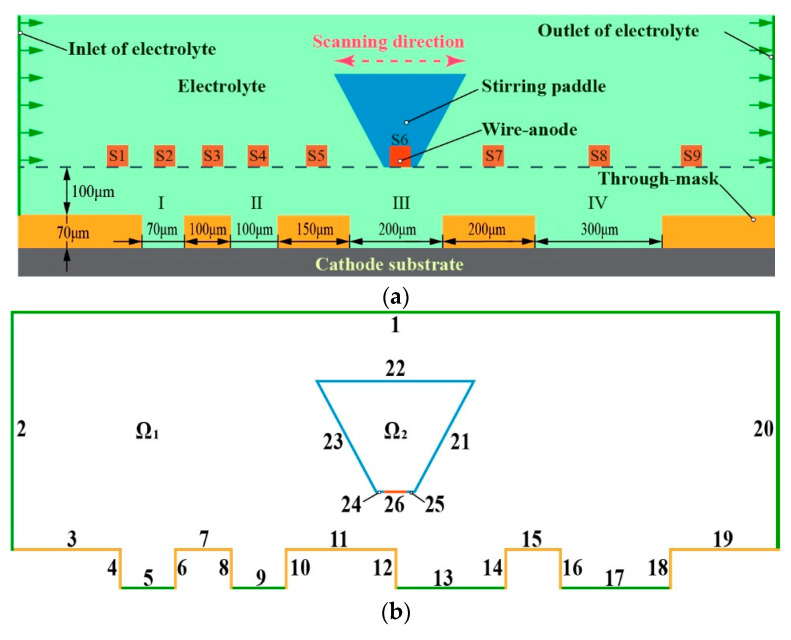
Schematic diagram of a simplified model of wire-anode scanning electroforming. (**a**) Diagram of the simplified geometric model. S1 to S9 are the wire-anodes located at 9 specific positions; (**b**) Diagram of domains and boundaries.

**Figure 5 micromachines-14-00979-f005:**
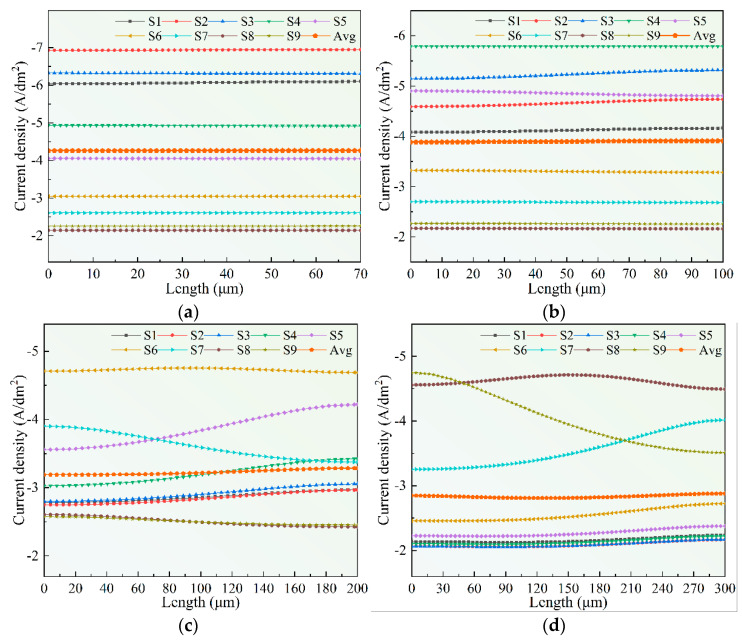
Diagram of current density distribution in each region when the anode is in different positions. (**a**) I; (**b**) II; (**c**) III; (**d**) IV.

**Figure 6 micromachines-14-00979-f006:**
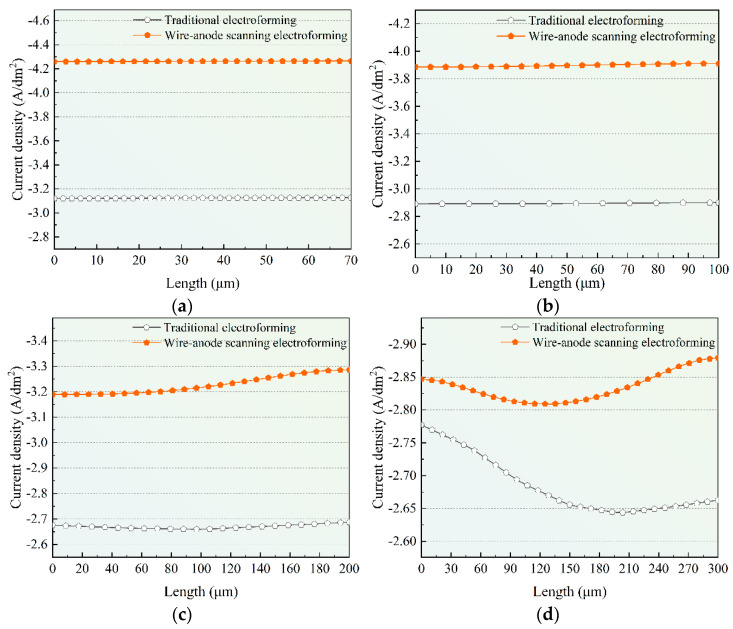
Schematic diagram of current density distribution in different areas under different electroforming methods. (**a**) I; (**b**) II; (**c**) III; (**d**) IV.

**Figure 7 micromachines-14-00979-f007:**
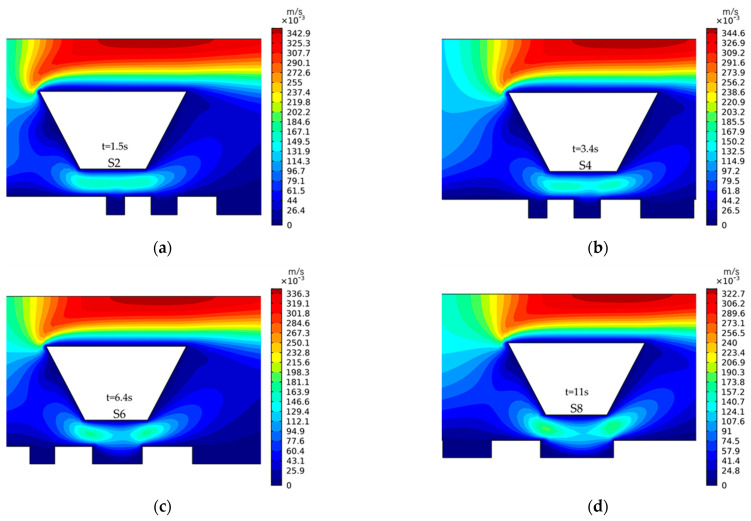
Diagram of flow field distribution in each region when the stirring paddle is directly above it. (**a**) I; (**b**) II; (**c**) III; (**d**) IV.

**Figure 8 micromachines-14-00979-f008:**
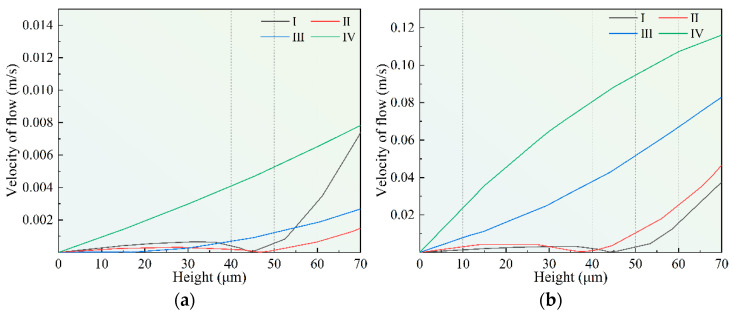
Diagram of the flow rate in each region of the stirring paddle under different states. (**a**) Static state; (**b**) When the stirring paddle is located directly above each area.

**Figure 9 micromachines-14-00979-f009:**
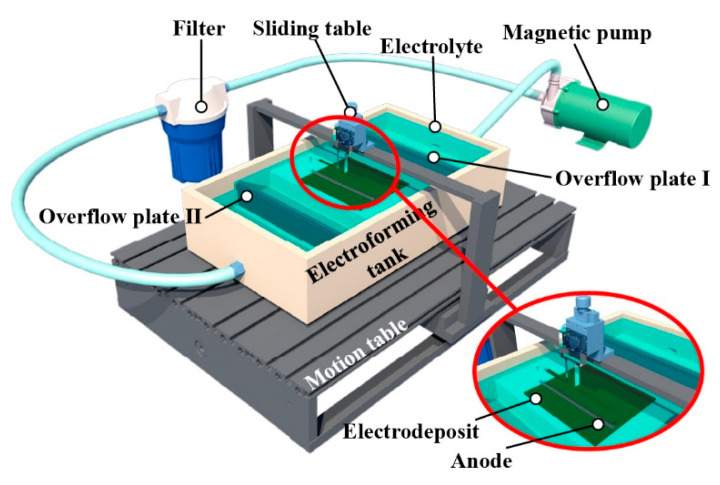
Schematic diagram of wire-anode scanning electroforming system.

**Figure 10 micromachines-14-00979-f010:**
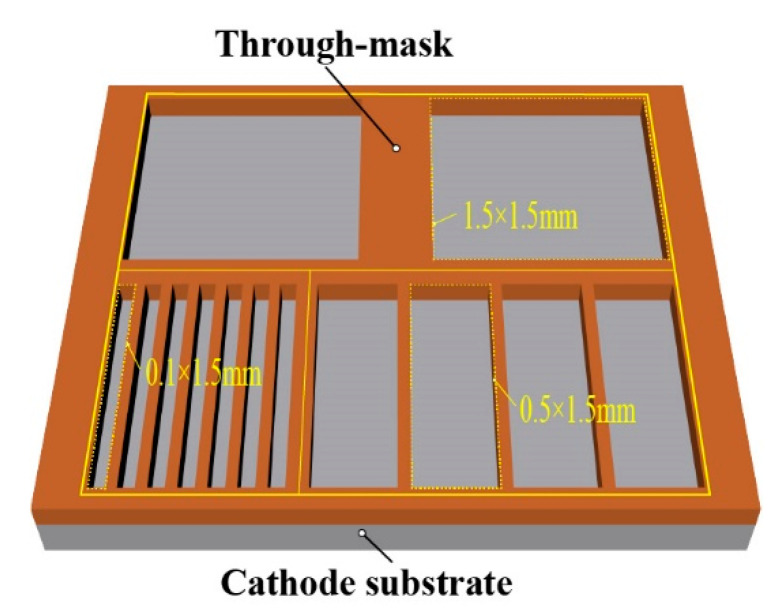
Schematic diagram of the designed multi-scale through-mask.

**Figure 11 micromachines-14-00979-f011:**
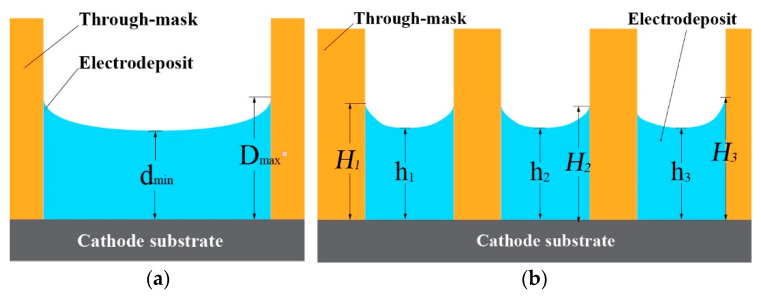
Schematic diagram of electroformed parts thickness evaluation. (**a**) single part; (**b**) part arrays.

**Figure 12 micromachines-14-00979-f012:**
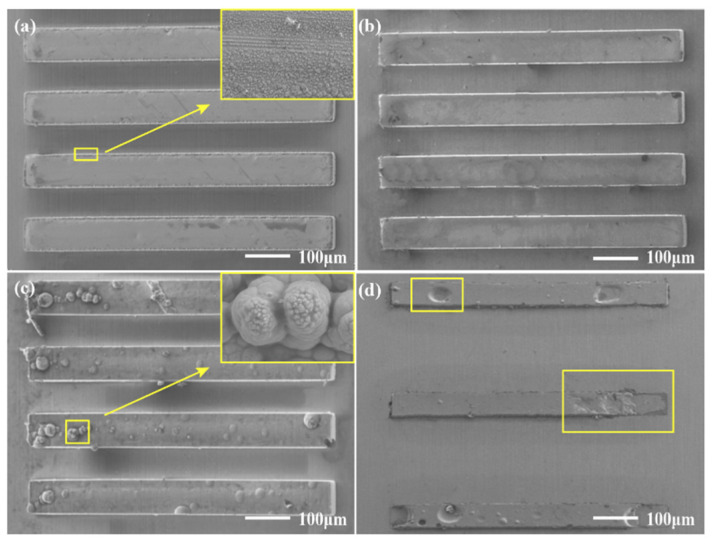
Surface morphologies of the electrodeposited 0.1 mm × 1.5 mm rectangular microscale features under different electroforming methods. (**a**) WAS-EF, 2.6 V; (**b**) WAS-EF, 4 V; (**c**) WAS-EF, 6 V; (**d**) Traditional electroforming.

**Figure 13 micromachines-14-00979-f013:**
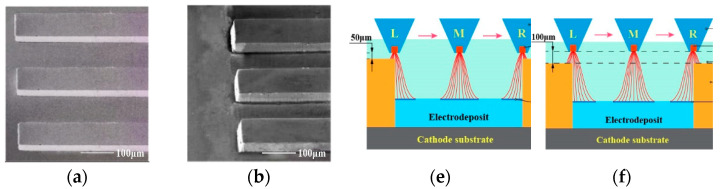
Profile and forming principle of wire-anode scanning electroforming with different working gaps. (**a**,**e**) 50 μm; (**b**,**f**) 100 μm; (**c**,**g**) 300 μm; (**d**,**h**) 600 μm.

**Figure 14 micromachines-14-00979-f014:**
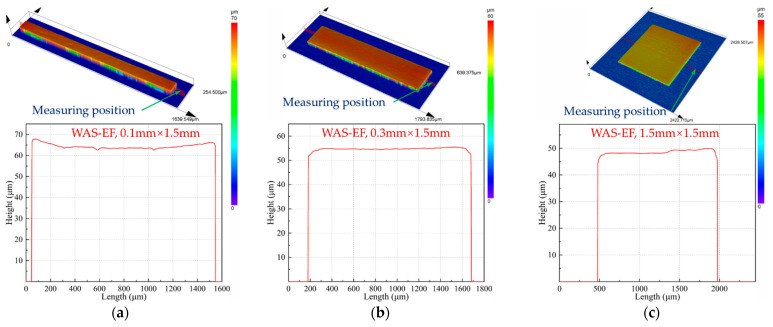
Geometric profile and thickness uniformity of the single part fabricated by different electroforming methods. (**a**) WAS-EF, 0.1 mm × 1.5 mm; (**b**) WAS-EF, 0.3 mm × 1.5 mm; (**c**) WAS-EF, 1.5 mm × 1.5 mm; (**d**) traditional TM-ECD, 0.1 mm × 1.5 mm; (**e**) traditional TM-ECD, 0.3 mm × 1.5 mm; (**f**) traditional TM-ECD, 1.5 mm × 1.5 mm.

**Figure 15 micromachines-14-00979-f015:**
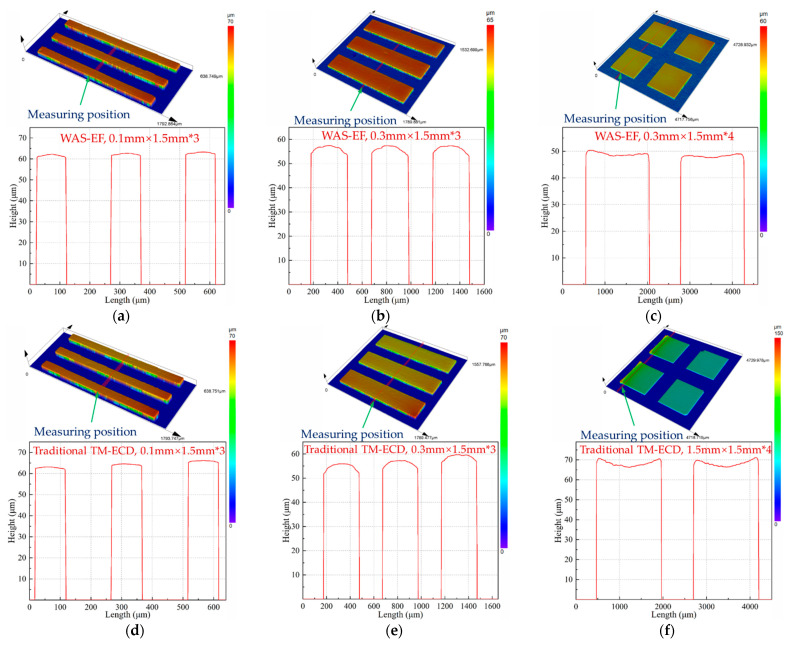
Geometric profile and thickness uniformity of the part arrays fabricated by different electroforming methods (**a**) WAS-EF, 0.1 mm × 1.5 mm; (**b**) WAS-EF, 0.3 mm × 1.5 mm; (**c**) WAS-EF, 1.5 mm × 1.5 mm; (**d**) traditional TM-ECD, 0.1 mm × 1.5 mm; (**e**) traditional TM-ECD, 0.3 mm × 1.5 mm; (**f**) traditional TM-ECD, 1.5 mm × 1.5 mm.

**Figure 16 micromachines-14-00979-f016:**
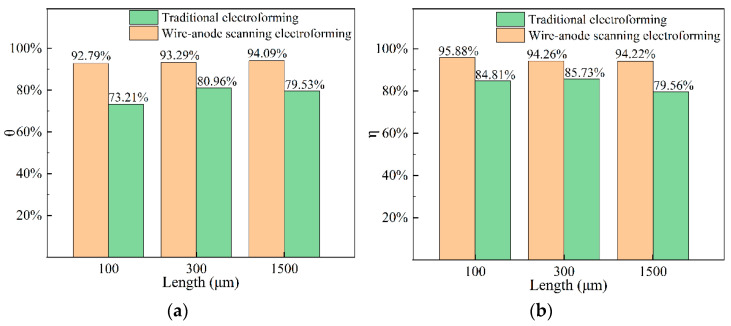
The value of θ and. (**a**) θ; (**b**) η.

**Table 1 micromachines-14-00979-t001:** Domain conditions of WAS-EF.

Domain Conditions	Domain	Property
Electrolyte	Ω_1_	ρ_1_ = 1.042 g/cm^3^; μ_1_ = 1.062 × 10^−3^ Pa·s σ_1_ = 10.3 S/m; T = 328.15 K
Stirring paddle	Ω_2_	ρ_2_ = 1.18 g/cm^3^; E = 3.16 × 109 Pa nu = 0.32
Specify deformation domain	Ω_2_	X = 0.00055 [m] × sin (2 × pi × t [1/s]); Y = 0

**Table 2 micromachines-14-00979-t002:** Boundary conditions for numerical simulation of wire-anode scanning electroforming.

Boundary Conditions	Boundary	Property
Electrolyte inlet	2	U_0_ = 0.1 m/s
Outlet	20	p_0_ = 0 Pa
Anode	26	I_0_ = 250 A/m^2^
Cathode	5, 9, 13, 17	0 V
Wall	3–19	Slip

## Data Availability

Not applicable.

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
