# Peer review of "Concurrently Fabricating Precision Meso- and Microscale Cross-Scale Arrayed Metal Features and Components by Using Wire-Anode Scanning Electroforming Technique"

_micromachines, 2023, doi:10.3390/mi14050979_

Round 1

Reviewer 1 Report

1. The introduction is too cumbersome and can be refined.

2. The entire article is simulating and experimenting with the TM-ECD mode. Is the Overgrowth-ECD mode necessary in Fig. 1(C)?

3. All the equations need reference from original source.

4. w in Eq. 1 should be indicated in Fig. 2(b)

5. Correct and recheck the subscript and superscript throughout the manuscript.

6. The manuscript lacks a description of the specific location of S1 to S9 in Fig. 4(a)

7. The manuscript mentions “depth to width ratio”, but only simulates structures of different sizes at the same height, and lacks simulation analysis of the same size at different heights. It cannot be ruled out that size size has an impact on uniformity. Therefore, the impact of depth to width ratio on uniformity cannot be mentioned.

8. Fig. 7 is not clear enough. Replace images with higher clarity to make the numbers visible

9. The simulation lacks explanations for parameters such as electroforming time and temperature, especially electroforming time, which cannot display the simulation results obtained at what electroforming time

10. In order to better compare traditional methods with proposed methods, merge Fig. 14 and Fig.15 into one figure and merge Fig. 16 and Fig.17 into one figure.

Author Response

Thank you for considering the revised version of our manuscript-Concurrently Fabricating Precision Meso- and Microscale Cross-scale Arrayed Metal Features and Components by Using Wire-anode Scanning Electroforming Technique. We are thankful to you for pointing out some important modifications needed in the report. Those comments are all valuable and very helpful for improving our manuscript, as well as the important guiding significance to our researches. We have carefully studied and reviewed the comments and have revised the manuscript accordingly which we hope meet with approval.

Reviewer 2 Report

The review concerns the manuscript titled "Concurrently Fabricating Precision Meso- and Microscale Cross-scale Arrayed Metal Features and Components by Using Wire-anode Scanning Electroforming Technique".

The manuscript in current form is not ready to publish in the Journal. Therefore, it should be supplemented according to the comments given below.

Please use the impersonal form in the body of the article.

Abstract. This paragraph should state the purpose of the research and the main results, preferably quantitatively.

I suggest introducing abbreviations and symbols at the beginning of the manuscript, and using them in the content.

Section 4.2. The Authors write about the surface topography but there are no results of this type of research. The manuscript shows the measurement results, but they are not accurate, the axis is missing, etc.
Please specify the parameters of topography and morphology measurements in this section .

The font in the drawings should be larger to be readable.

Author Response

(The authors gave the same response as above.)

Reviewer 3 Report

This work offers a novel TM-ECD method that differs fundamentally from previous TM-ECD processes by employing a sizeable surface-area ratio between the anode and cathode and a large IEG. A linear wire-shaped ultrafine (diameter/width 50m) electrically inert anode (abbreviated as wire-anode) attached on a reciprocating paddle is employed in the proposed TM-ECD, and the working gap between the anode and the top surface of the through-mask is maintained below 500 m. It is predicted that WAS-EF will enable high-precision deposition of a range of multi-scale components on the same cathode simultaneously. The work introduces a fabrication process with great potential to commercialize precision cross-scale arrayed metal components via wire-anode scanning electroforming technique. Still, a few questions should be addressed before publication.

1. This work presents complete and sufficient simulation data and a nice schematic diagram to elaborate the concept of this work, which is very meaningful to contribute, but in my opinion, the experimental data is somehow not enough to support the idea from the schematic diagram or simulation. For example, Fig. 3 (a) shows many different shapes of electrodeposited metal. Still, only long columnar metals SEM data were presented in Fig. 12 and Fig. 13. The whole work is more like a simulation paper, not a practical fabrication process development work.  

2. The legend numbers in Fig. 5 and the % numbers in Fig. 18 are too small to see; please enlarge them.

3. Authors observed nodules on the surface of the parts of the metal component. Have the Authors done any component analysis of those nodules?

A few minor grammars issues in the manuscript; please check the usage of the definite article.

Author Response

(The authors gave the same response as above.)

Round 2

Reviewer 2 Report

I do not have more comments.

Thank you.